# Inference of Sequential Patterns for Neural Message Passing in Temporal Graphs

## Abstract

The modelling of temporal patterns in dynamic graphs is an important current research issue in the development of time-aware Graph Neural Networks (GNNs). However, whether or not a specific sequence of events in a temporal graph constitutes a *temporal* pattern not only depends on the frequency of its occurrence. We must also consider whether it deviates from what is expected in a temporal graph where timestamps are randomly shuffled. While accounting for such a random baseline is important to model temporal patterns, it has mostly been ignored by current temporal graph neural networks. To address this issue we propose HYPA-DBGNN, a novel two-step approach that combines (i) the inference of anomalous sequential patterns in time series data on graphs based on a statistically principled null model, with (ii) a neural message passing approach that utilizes a higher-order De Bruijn graph whose edges capture overrepresented sequential patterns. Our method leverages hypergeometric graph ensembles to identify anomalous edges within both first- and higher-order De Bruijn graphs, which encode the temporal ordering of events. Consequently, the model introduces an inductive bias that enhances model interpretability.

We evaluate our approach for static node classification using established benchmark datasets and a synthetic dataset that showcases its ability to incorporate the observed inductive bias regarding over- and under-represented temporal edges. Furthermore, we demonstrate the framework's effectiveness in detecting similar patterns within empirical datasets, resulting in superior performance compared to baseline methods in node classification tasks. To the best of our knowledge, our work is the first to introduce statistically informed GNNs that leverage temporal and causal sequence anomalies. HYPA-DBGNN represents a promising path for bridging the gap between statistical graph inference and neural graph representation learning, with potential applications to static GNNs.

## 1 Introduction

Graphs are powerful representations of complex data. Not surprisingly, there is a growing collection of successful methods for learning on graphs (Kipf & Welling, 2017; Veličković et al., 2018; Hamilton et al., 2018). These methods are versatile and are widely used in bioinformatics (Zhang et al., 2021), social sciences (Phan et al., 2023), and pharmacy (Stokes et al., 2020). While many methods assume a static graph, real-world scenarios often involve dynamic systems, such as evolving interactions in social networks. Although known techniques for static graphs can be applied to dynamic graphs (Liben-Nowell & Kleinberg, 2007), important patterns may be missed (Xu et al., 2020). Recently, several approaches have incorporated temporal dynamics to obtain time-aware graph neural networks. These methods are applied to different tasks such as static node classification (Qarkaxhija et al., 2022), link prediction or continuous node property prediction (Rossi et al., 2020). In this work, we focus on static node classification for graphs where patterns are encoded in the temporal order of edge activation. The key idea is that this temporal order crucially influences the role of nodes, which can be leveraged for static node classification. As an example, in a network of interactions between employees in a company, the specific temporal order in

which actors interact with their peers may depend on their role. Hence, two actors that are indistinguishable based on the static topology may still exhibit different temporal interaction patterns.

A common theme between static and temporal GNNs is that the observed graphs are usually directly used for message passing. Recently data augmentation techniques have been proposed to improve the generalizability of GNNs. Such data augmentation techniques have been considered for a variety of reasons such as to reduce oversquashing (Topping et al., 2021), improve class homophily for node classification (Liu et al., 2022), foster diffusion (Zhao et al., 2021a), or include non-dyadic relation-ships (Qarkaxhija et al., 2022). Another motivation that has recently been highlighted by Zhao et al. (2021c) is the presence of noise in empirically observed graphs. This motivates augmentation techniques for GNNs that ideally prune spuriously observed edges, while adding erroneously unobserved edges.

However, addressing noise in observed graphs arguably requires *graph correction* methods accounting for a "random baseline" that allows to distinguish significant patterns from noise, rather than *augmentation* methods that are based on heuristics or adjust the graph based on ground truth node classes. Moreover, the application of GNNs to temporal graphs introduces unique challenges for data augmentation as we typically want to focus on temporal patterns that are due to the time-ordered sequence of events. To the best of our knowledge, no existing works have considered graph correction methods that combine a statistically principled inference of sequential patterns with temporal GNNs.

Addressing this research gap, in this work we propose HYPA-DBGNN, a novel two-step approach for temporal graph learning: In a first step we infer anomalous sequential patterns in time series data on graphs based on a *statistical ensemble* of temporal graphs, i.e. a null model of random graphs that preserves the frequency of time-stamped edges but randomizes the temporal ordering in which those edges occur. Building on the HYPA framework (LaRock et al., 2020), our method leverages hypergeometric graph ensembles. This allows us to analytically calculate expected frequencies of node sequences on time-respecting paths, which is the basis to identify anomalous sequential patterns in temporal graphs. Consider the interaction sequence $\langle AXC \rangle$ in Figure 1. Compared to the other observed sequences, it has a low frequency and thus it would have a low impact in the computations of a standard GNN. However, accounting for the frequencies of its sub-sequences reveals that it occurred more often than expected, i.e., it is *overrepresented*. Discarding this information can over- or underplay the role of an interaction, thus negatively affecting results. Therefore, in the second step of our approach, we apply neural message passing on an augmented higher-order De Bruijn graph, whose edges capture overrepresented sequential patterns in a temporal graph. This introduces an inductive bias that emphasizes *sequential patterns* over mere edge frequencies. The contributions of our work are as follows:

(i) We propose a novel approach to augment message passing based on a statistical null model. This allows us to infer which temporal sequences in a time-stamped interaction sequence are over- or under-represented compared to a random baseline temporal graph in which the frequency of edges are preserved while their temporal ordering is shuffled.

(ii) Building on this statistical inference approach, we propose HYPA-DBGNN, a time-aware temporal graph neural network architecture that specifically captures temporal patterns that deviate from a random baseline.

(iii) We demonstrate our approach in synthetic temporal graphs sampled from a model that generates heterogeneously distributed temporal sequences of events in such a way that node classes are associated with the over- or underrepresentation of temporal events compared to random temporal orderings rather than mere frequencies.

(iv) We demonstrate the practical relevance of our method by evaluating node classification in five empirical temporal graphs capturing time-stamped proximity events between humans. A comparison of HYPA-DBGNN with standard De Bruijn Graph Neural Networks without our HYPA-based inference reveals that our approach yields an improved accuracy in all five data sets. Moreover, a comparison to seven baseline techniques shows that our method yields the best performance in all empirical data.

(v) We finally show that the distribution of HYPA scores in the augmented message passing graph, which captures the degree to which frequencies of temporal sequences

deviate from a random baseline, enables us to explain why HYPA-DBGNN yields larger performance improvements on some data sets compared to others.

Different from prior works, we propose a *statistically principled* data augmentation for temporal graph neural networks that uses a *statistical ensemble* of temporal graphs with a given weighted topology. Besides improving temporal GNNs, we further argue the general approach of utilizing well-known *statistical ensembles of graphs* from network science for *graph correction* could help to improve the performance of GNNs in data affected by noise.

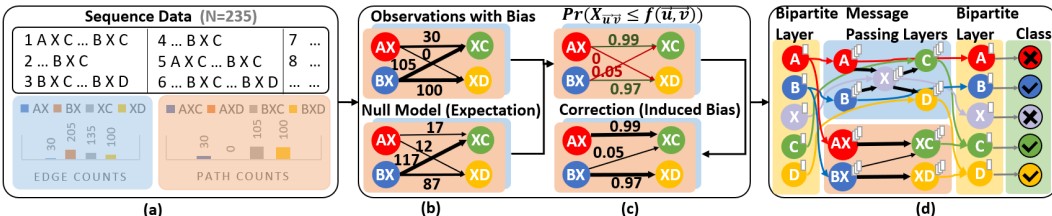

Figure 1: Inference procedure leading to the dynamic graph used for neural message passing. (a) Example of sequence data adapted from LaRock et al. (2020). (b) First- (blue) and higher-order (orange) De Bruijn graphs encoding temporal ordered time-stamped edges are compared to random graph ensemble null model with shuffled time-stamped k-1-order edges. (c) The graphs are corrected by introducing a statistical-principled bias that revalues all edges ($w_{\langle AXC \rangle} \approx w_{\langle BXD \rangle} > w_{\langle BXC \rangle}$) and removes under-represented edges, i.e. edges that appear with a high probability less than expected ($\langle AXD \rangle$). (d) The multi-order graph neural network is trained respecting the inferred graphs.

## 2 Related Work

Data augmentation for graphs has been explored from various directions with the goal of allowing machine learning models to better generalize and attend to signal over noise (Zhao et al., 2022). Many methods have utilized heuristic graph modification strategies like randomly removing nodes (Feng et al., 2020), edges (Rong et al., 2019), or subgraphs (Wang et al., 2020; You et al., 2020) to improve performance and generalizability. Other works have considered adding virtual nodes (Pham et al., 2017; Hwang et al., 2021) or rewiring the network topology, which also addresses oversquashing (Topping et al., 2021; Barbero et al., 2023), with graph transformers operating on a fully connected topology representing an extreme case (Mialon et al., 2021; Ying et al., 2021; Kreuzer et al., 2021). Additionally, it has been shown that using graph diffusion convolutions instead of raw neighborhoods alleviates problems from noisy and arbitrarily defined edges in real-world graphs (Gasteiger et al., 2019). Network data augmentation has also been explored by going beyond pairwise connections, either through mediating node interactions via subgraphs (Monti et al., 2018; Bevilacqua et al., 2021; Zhao et al., 2021b; Cotta et al., 2021) or by utilizing higher-order graphs. Examples of higher-order approaches include simplicial networks (Bodnar et al., 2021b), cellular complexes (Bodnar et al., 2021a; Hajij et al., 2022), hypergraphs (Huang & Yang, 2021; Chien et al., 2021; Georgiev et al., 2022), and time-respecting node sequences (Qarkaxhija et al., 2022). Another area of research focused on learning the graph augmentations from the data. One approach is to perform graph augmentation as a preprocessing step, completely separate from the downstream task, where the graph structure is cleaned before being used as input to the GNN (Jin et al., 2020; Zhao et al., 2021c). Other works embed the augmentation strategy into an end-to-end differentiable pipeline, jointly learning the optimal graph representation and the downstream task (Jiang et al., 2019; Lu et al., 2024; Franceschi et al., 2019; Fatemi et al., 2021; Kazi et al., 2022).

As our work addresses temporal graph data, it is related to the field of temporal GNNs. Temporal GNNs have been developed for both discrete- and continuous-time settings (Longa et al., 2023). Discrete-time approaches segment the temporal data into time windows (Liben-Nowell & Kleinberg, 2007; Sankar et al., 2020; Hajiramezanali et al., 2019), thus aggregating interactions and losing information on time-respecting paths within those time windows. In

contrast to the discrete-time setting, continuous-time approaches produce time-evolving node embeddings, focusing on the temporal variability of network activity at different time points, rather than on the patterns occurring across temporally-ordered interaction sequences (Xu et al., 2020; Rossi et al., 2020; Kumar et al., 2019). These methods are commonly evaluated based on the prediction of dynamically changing node labels, which differs from the prediction of static node labels with sequential information that we consider in our work. The work most similar to our perspective is DBGNN (Qarkaxhija et al., 2022), which learns from sequential correlations in high-resolution timestamped data. Our approach diverges from DBGNN by considering a more nuanced notion of the relevance of time-respecting paths that involves structural changes to the graph. Rather than relying on the raw frequency of interactions, HYPA-DBGNN uses a statistically grounded anomaly score. This score quantifies the over- and under-expression of time-respecting paths, making the model less susceptible to noise while basing contribution of paths on their statistical significance.

## 3 BACKGROUND

A graph $G = (V, E)$ is defined as a set of nodes $V$ representing the elements of the system, and a set of edges $E \subseteq V \times V$ representing their direct connections. However, it is often important to consider how nodes influence one another through a *path*, which is an ordered sequence $(v_0, v_1, \ldots, v_l)$ of nodes $v_i \in V$. In a path, all node transitions must correspond to edges in the graph, i.e., $e_i = (v_i, v_{i+1}) \in E; \forall i \in [0, l-1]$. Paths are often inferred from edges based on a *transitivity assumption*. This assumption states that if there is an edge $(v_0, v_1)$ with transition probability $\alpha$, and an edge $(v_1, v_2)$ with transition probability $\beta$, then the path $(v_0, v_1, v_2)$ will be observed with probability $\alpha \cdot \beta$. In other words, the transitions are considered to be independent. The transitivity assumption simplifies the modeling of a path by expressing its probability as the product of the individual edge transition probabilities. However, this assumption often fails in temporal networks $G^t = (V, E^t)$, where $E^t \subseteq V \times V \times \mathbb{N}$ as edges have timestamps. In temporal networks, the ordering of edges can play an important role in determining the likelihood of observing certain paths. A *time-respecting* path is defined as a sequence of edges $((v_0, v_1, t_1), \ldots, (v_i, v_{i+1}, t_{i+1}), \ldots, (v_{l-1}, v_l, t_l))$ that $\forall i \in [0, l-1]$ respects two conditions: (i) transitions respect the order of time $t_i > t_{i-1}$, and (ii) $t_i - t_{i-1} \leq \delta$, where $\delta$ is a parameter controlling the maximum time distance for considering interactions temporally adjacent. Therefore, different from what we would get by discarding time and using the transitivity assumption, the two edges $(v, w, t_1)$ and $(u, v, t_2)$ form a time-respecting path only if $t_2 > t_1$. To capture time-respecting sequential patterns, higher-order De Bruijn graphs model the probabilities of path sequences explicitly. These models construct a representation that respects the topology of the original graph and the frequencies of observed paths of a given length $k$. Specifically, a higher-order network of the k-th order is defined as an ordered pair $G^{(k)} = (V^{(k)}, E^{(k)})$, where $V^{(k)} \subseteq V^k$ are the higher-order vertices, and $E^{(k)} \subseteq V^{(k)} \times V^{(k)}$ are the higher-order edges. $V^k$ contains all k-th order vertices that exists as paths in the first-order graph $G$ whereas $V^{(k)}$ contains the subset of k-th order vertices that exist as path in the observed data. Each higher-order vertex $v =: \langle v_0 v_1 \ldots v_{k-1} \rangle \in V^{(k)}$ is an ordered tuple of $k$ vertices $v_i \in V$ from the original graph. The higher-order edges connect higher-order nodes that overlap in exactly $k - 1$ vertices, similar to the construction of high-dimensional De Bruijn graphs (De Bruijn, 1946). The weights of the higher-order edges in $G^{(k)}$ represent the frequency of paths of length $k$ in the original graph. Specifically, the weight of the edge $(\langle v_0 \ldots v_{k-2} \rangle, \langle v_1 \ldots v_{k-1} \rangle)$ counts how often the path $\langle v_0 \ldots v_{k-1} \rangle$ of length $k$ occurs. By explicitly modeling the probabilities of these higher-order path sequences, the higher-order network representation can capture patterns and dependencies that may be missed when relying on the transitivity assumption (Scholtes, 2017).

**Detection of Path Anomalies** Defining anomalies requires a reference base. In our case, the transitivity assumption provides the null model that serves as this baseline. Anomalies occur in sequences that deviate from this baseline, likely due to correlations and interdependencies not captured by the transitivity assumption. First, we discuss how the hypergeometric ensemble allows testing for anomalous edge frequencies based on node activity, i.e., their in- and out-degrees. Building on this, we then outline how this methodology is extended to test if the frequencies of paths of length $k$ are anomalous given those of paths of length $k - 1$.

Configuration models (Molloy & Reed, 1995) provide randomization methods for graphs that shuffle edges while preserving vertex degrees. In a nutshell, first, they disassemble the graph, leaving nodes with in- an out-stubs. Then, a new network is reassembled by connecting pairs of in- and out- are picked with equal probability. This procedure is algorithmically straightforward but can be computationally expensive. To address this, Casiraghi & Nanumyan (2021) contributed a closed-form expression for the *soft* configuration model, which fixes the *expected* vertex degrees rather than the exact degree sequence. In their formulation, the sampling of edges is equated to sampling from an urn. The authors introduce a combinatorial matrix $\mathbf{\Xi} \in \mathbb{N}^n \times \mathbb{N}^n$, where $\Xi_{ij} = d_i^{out} \cdot d_j^{in}$ encodes the product of the out-degree of node $i$ and the in-degree of node $j$ in the original graph $G$. The total number of possible edge placements is then $M = \sum_{ij} \Xi_{ij}$. A network is sampled from this ensemble by drawing $m = \sum_i d_i^{out} = \sum_i d_i^{in}$ edges without replacement from the $M$ possible edge placements. The probability of observing $A_{ij}$ edges between nodes $i$ and $j$ is then given by the hypergeometric distribution: $P(A_{ij}) = \binom{M}{m}^{-1} \binom{\Xi_{ij}}{A_{ij}} \binom{M-\Xi_{ij}}{m-A_{ij}}$. Having this probability mass function, we can use the equation above to quantify the anomalousness of the frequency of an edge. This closed-form expression and the sampling process that generates it provides a principled null model that preserves the expected degree sequence, which will be crucial for our subsequent analysis of anomalous path patterns in the network.

Our concept of path anomalies, introduced by LaRock et al. (2020), provides a statistical framework for identifying paths through a graph that are traversed with anomalous frequencies. The key idea is to define a null model of order $k - 1$ that captures the expected frequencies of paths of length $k$, and then identify paths that deviate significantly from this null model. To construct the null model, one must establish a statistical ensemble of $k$-th order De Bruijn graphs. The starting point is the hypergeometric ensemble outlined in the previous paragraph, which preserves the total in- and out-degrees of nodes while shuffling the edge weights. For a De Bruijn graph of order $k$, the nodes' degrees are determined by the frequencies of paths of length $k - 1$, i.e., by the edge frequencies of De Bruijn graph of order $k - 1$. A hypergeometric ensemble of the De Bruijn graph presents one additional difficulty. Specifically, an edge between two k-th order nodes is valid only if their path representations overlap in $k - 1$ first-order nodes. This implies that some of the $\mathbf{\Xi}$ matrix entries represent invalid paths. HYPA handles this by zeroing out impossible entries and redistributing their values through an optimization procedure, as detailed in the original work.

## 4    HYPA De Bruijn Graph Neural Network Architecture

We now introduce the HYPA-DBGNN architecture [1] that relies on statistical-principled graph augmentation. The temporal dynamics of the sequential patterns are encoded in first- and higher-order De Bruijn graphs. Graph corrections are inferred that include anomaly statistics in the graph topology. We then present a multi-order augmented message passing scheme that relies on the inferred graphs with induced bias. Although we adapt the message passing procedure of Graph Convolution Networks (GCN) from Kipf & Welling (2017), our architecture is generalizable to other message passing schemes due to the selective additions.

**Statistical-Principled Graph Augmentation**    As outlined before, the k-th-order De Bruijn graphs capture the observed frequencies of the k-th-order sequences through the edges between k-th-order nodes. This potentially biased representation yields the foundation for hypergeometric ensembles whose edge frequencies are induced by the (k-1)-th-order sub-sequences. The HYPA score (LaRock et al., 2020), defined as $HYPA^{(k)}(u,v) = \Pr(X_{uv} \leq f(u,v))$, uses these to describe how probable an observed edge has a higher frequency than in any random realization. A large HYPA score encodes edges that are observed more than expected whereas a HYPA score approaching zero describes edges that are observed less than expected. Leveraging the HYPA scores as adjacency matrix $A_{uv}^{(k)} = HYPA^{(k)}(u,v)$ leads to corrected graphs where the weights of underrepresented edges are reduced and the weights of overrepresented edges are scaled based on the expected value. To improve the scalability of our approach we preserve the sparsity of the observed higher-order graph by

[1]A reference implementation, data sets and benchmarks are given at [blinded].

not including HYPA scores of unobserved edges while adding HYPA scores of observed edges as edge attributes.

**Message Passing for Higher-Order De Bruijn Graphs with Induced Bias**    For layer $l$, we define the update rule of the message passing as

$$\vec{h}_v^{k,l} = \sigma \left( \mathbf{W}^{k,l} \sum_{\{u \in V^{(k)}:(u,v) \in E^{(k)}\} \cup \{v\}} \frac{HYPA^{(k)}(u,v) \cdot \vec{h}_u^{k,l-1}}{\sqrt{H(v) \cdot H(u)}} \right), \qquad (1)$$

with the previous hidden representation $\vec{h}_u^{k,l-1}$ of node $u \in V^{(k)}$, the inferred HYPA score $HYPA^{(k)}(u,v)$ of the given edge $(u,v) \in E^{(k)}$ (capturing the induced bias), the trainable weight matrices $\mathbf{W}^{k,l} \in \mathbb{R}^{H^l \times H^{l-1}}$, the normalization factor based on the HYPA score sum of incoming edges $H(v) := \sum_{\{u \in V^{(k)}:(u,v) \in E^{(k)}\} \cup \{v\}} HYPA^{(k)}(u,v)$, and the non-linear activation function $\sigma$, here ReLU. We want to highlight that integrating the inferred scores is a major technical challenge with room for further studies. We compare different variants in Appendix A and Appendix B.

Depending on order $k$, the message passing for different higher order graphs is based on different higher order node sets $V^{(k)}$ whose nodes $v$ have their own hidden representations $\vec{h}_v^{k,l}$ for every layer $l$. An initial feature encoding is only provided for the first order ($k = 1$) as $\vec{h}_v^{1,0}$. To transfer the features to higher-order nodes and to merge the hidden representations, we introduce two bipartite mappings.

The initial first-order feature set $\vec{h}_u^{1,0}$ is mapped to the higher-order node representations $\vec{h}_v^{k,1}$ using the bipartite graph $G^{b_0} = \left( V^{(k)} \cup V, E^{b_0} \subseteq V \times V^{(k)} \right)$ with $e_{uv} \in E^{b_0}$ if $v = (v_0, \ldots, v_{k-1}) \in V^{(k)}$ and $v_0 = u$ in analogy to interpreting message passing layers as higher-order Markov chains. This advancement enables the propagation of features to the higher-order graph. Multiple first-order representations are aggregated using the function $\mathcal{F}$ (in our case MEAN) and transformed with the learnable weight matrix $\mathbf{W}^{b_0} \in \mathbb{R}^{H^{1,0} \times H^{k,0}}$ to the higher-order feature space.

$$\vec{h}_v^{k,1} = \sigma \left( \mathbf{W}^{b_0} \mathcal{F} \left( \left\{ \vec{h}_u^{1,0} : \text{for } u \in V^{(1)} \text{ with } (u,v) \in E^{b_0} \right\} \right) \right) \qquad (2)$$

The second mapping is defined as by Qarkaxhija et al. (2022). It is the counterpart to the first bipartite layer. Here, the higher-order node representations are summed with the first-order node representations (requiring matching representation dimensions $F^g = H^l$) if the *last* path entry $u_{k-1}$ of the higher-order node $u = (u_0, \ldots, u_{k-1}) \in V^{(k)}$ equals the first-order node $v = u_{k-1}$. The bipartite graph is given as $G^b = \left( V^{(k)} \cup V, E^b \subseteq V^{(k)} \times V \right)$ and leads to a first-order node representation $\vec{h}_v^b$ with $v \in V$ and the learnable matrix $\mathbf{W}^b \in \mathbb{R}^{F^g \times H^l}$.

$$\vec{h}_v^b = \sigma \left( \mathbf{W}^b \mathcal{F} \left( \left\{ \vec{h}_u^{k,l} + \vec{h}_v^{1,g} : \text{for } u \in V^{(k)} \text{ with } (u,v) \in E^b \right\} \right) \right) \qquad (3)$$

In Figure 1 we show an overview of the inference process and the proposed neural network architecture for the first- and second-order graph. Moreover, Appendix C contains a more detailed visualization of the architecture. We rely on a one-hot encoding as first-order feature set. The fist bipartite layer transfers this to the higher-order nodes. The neural network performs multiple message passing steps independently for the two given graph topologies. The number of message passing rounds and the dimensions of layers may vary in the two parts. After performing the message passing in parallel and merging the features with the second bipartite layer a final classification layer is applied. The model architecture allows to include node and edge features due to the underlying GCN. The computational complexity of HYPA-DBGNN is upper-bounded by the complexity of the baseline DBGNN due to the edges removed in graph correction.

For a more detailed discussion on the computational complexity we refer the reader to Appendix D. We note that the computational complexity is not a limiting factor for our approach. This is supported by the event count $2^{23}$ in the synthetic data set that is

comparable to the events in small to medium size TGB data sets Huang et al. (2023) or to the edge counts in corresponding non-temporal graphs in OGB data sets (Hu et al., 2020). We also report the overall needed training resources (Appendix E) for the used data sets together with their properties (Appendix F).

## 5 EXPERIMENTAL EVALUATION

We compare our architecture with graph representation learning methods (**EVO** (Belth et al., 2020), **HONEM** (Saebi et al., 2020), **DeepWalk** (Perozzi et al., 2014) and **Node2Vec** (Grover & Leskovec, 2016)) and deep graph learning methods (**GCN** (Kipf & Welling, 2017), **LGNN** (Chen et al., 2020) and **DBGNN** (Qarkaxhija et al., 2022)). Finally, we also consider the state of the art dynamic node prediction method **TGN** (Rossi et al., 2020). This method was developed for predicting changes of nodes labels over time, and not for the prediction of static node labels that depend on the sequences of interactions. Therefore, we adapt the original training procedure to fit the static task as outlined in Appendix G. For the representation learning models Node2Vec and EVO we adhere to the original configurations, i.e. we use an embedding size of $d = 128$ and a random walk length of $l = 80$, repeated $r = 10$ times. As context size we use $k = 10$. For Node2Vec we select the return parameter ($p$) and the in-out parameter ($q$) from the set $0.25, 0.5, 1, 2, 4$. The deep learning models (GCN, LGNN, DBGNN, and our proposed model) consist of three layers. Following the approach of Qarkaxhija et al. (2022), we set the size of the last layer to $h_2 = 16$, while the sizes of the preceding layers are determined during model selection. The study range for $h_0$ and $h_1$ encompasses $4, 8, 16, 32$ over a maximum of 5000 epochs as per Qarkaxhija et al. (2022). The higher-order path length is fixed to $k = 2$ for HYPA-DBGNN and DBGNN because it is shown as optimal by Qarkaxhija et al. (2022) for the given data sets. Stochastic Gradient Descent (SGD) serves as our optimization function, with the learning rate set to 0.001, which showed the best performances. We use dropout regularization with a dropout rate of 0.4 to mitigate overfitting and we incorporate class weights in the loss function to address imbalanced training datasets.

To compare various Graph Neural Network (GNN) architectures, we adopt a conventional approach as documented in literature (Errica et al., 2019; Morris et al., 2020; James et al., 2013). For the assessment of model generalizability, we employ a nested cross-validation strategy with $N = 10$ repetitions. The data undergoes stratified partitioning into nine training and one testing fold, further divided into stratified training and validation subsets (80/20%) within each repetition. Subsequently, we select the best-performing model and epoch based on its validation set performance. Finally, we evaluate the chosen model's performance on the test set, reporting the mean and standard deviation of the respective metric across all N repetitions. For comparability, we use the same folds and splits for all experiments. Besides the random splits, the random initialization of the model also contributes to the variability captured by the standard deviation. For reproducibility, we fix the random splits and reuse a common seed in every repetition for the random initialization of model weights and dropout candidates.

We additionally perform two ablation studies to investigate different aspects of our method. First, Appendix H shows the contribution of both parts of the novel combination of statistically inference with machine learning by disabling the individual components of our architecture. Second, finding a suitable way of integrating the statistical inferred information is a key technical challenge with potential for further work. In Appendix A we present different integrations based on the Z-score for even more efficiency and numerical stability and a pruning-based approach. However, the approach we evaluate in the following leads to the best performance, as shown in Appendix B.

### 5.1 EXPERIMENTAL RESULTS FOR SYNTHETIC DATA SETS

We use synthetic data with two classes of nodes $C = \{A, B\}$ to demonstrate the type of patterns that only our model can learn. The characteristic properties and its derivation of the configuration model are detailed in Appendix I. Importantly, it contains a heterogeneous sequence (e.g. $\langle v_0, v_1, v_2 \rangle_f$) distribution of time-stamped edges or events (here:

$(v_0, v_1)_{t_0}, (v_1, v_2)_{t_1}, t_0 < t_1)$ between nodes. The learnable sequential pattern is an increased class-assortativity, i.e. edges are temporally ordered such that same class events are preferred followed by each other (e.g. leading to $\langle A, A, A, B \rangle$). Hence, these higher-order sequences with nodes from the same group are over-expressed compared to what we would expect by shuffling the temporal-order of the timestamped-edges between the nodes (e.g. $\langle A, B, A, A \rangle$).

The pattern is only discernible by higher-order models due to its restriction to higher-order sequences. For a homogeneous sequence distribution the pattern of overrepresented sequences would be reflected in the mere frequencies. However, due to the initial heterogeneous distribution, overrepresented sequences can also have low frequencies (e.g. $\langle A, X, C \rangle$ in Figure 1). Thus, they are also unobservable by higher-order baselines only including mere frequencies like DBGNN. However, the comparison of the observed frequencies with a null model that preserves the frequency of time stamped-edges but randomizes the temporal order reveals the sequential pattern.

We use two synthetic data sets with the same distribution of time-stamped edges. *Unweighted Sampling* contains sequences with randomized temporal order of time-stamped edges whereas *Weighted Sampling* contains sequences with increased class-assortativity. An unintended correlation between the obtained graph topology and the event classes is not excluded for both data sets. *Weighted Sampling* additionally contains the preferential chaining pattern.

The results in Table 1 show the capabilities of the models in terms of accuracy in solving the respective binary node classification tasks. The different methods yield varying results for synthetic data set without intended pattern. All representation learning methods with a horizon of $l = 80$, except EVO, perform better than the deep graph learning methods with a smaller horizon of $l = 3$. Our approach performs as good as DBGNN that shares similarities, like two distinct message passing modules, in its architecture.

The *Weighted Sampling* highlights the ability of the methods to learn the intended pattern. For the second data set GCN performs worse and all other baselines methods perform equal as for the first one. In contrast, HYPA-DBGNN improves by 55% and reaches an accuracy of 100%. These observations lead to the result that some of current baselines are able to learn an unintended pattern in both data sets. However, they fail in learning the implanted increased class-assortativity pattern whereas HYPA-DBGNN is able to learn this pattern.

Table 1: Comparison of HYPA-DBGNN baselines for the synthetic data sets. The table presents the balanced accuracy and its standard deviation for the static node classification task on dynamic graphs as obtained through the outlined experiments. The *Unweighted Sampling* data set contains a heterogeneous sequence distribution of time-stamped edges with shuffled temporal order. The adapted distribution of sequences in *Weighted Sampling* encodes a sequential pattern such that time-stamped edges between nodes of the same class are overrepresented but not necessarily very frequent.

| Representation Learning | EVO | HONEM | DeepWalk | Node2Vec | |
|---|---|---|---|---|---|
| Unweighted Sampling | $40.00_{\pm 31.62}$ | $80.00_{\pm 25.82}$ | $60.00_{\pm 21.08}$ | $60.00_{\pm 21.08}$ | |
| Weighted Sampling | $40.00_{\pm 31.62}$ | $80.00_{\pm 25.82}$ | $60.00_{\pm 21.08}$ | $60.00_{\pm 21.08}$ | |

| Deep Graph Learning | GCN | LGNN | DBGNN | TGN | HYPA-DBGNN |
|---|---|---|---|---|---|
| Unweighted Sampling | $50.00_{\pm 33.33}$ | $50.00_{\pm 0.00}$ | $45.00_{\pm 28.38}$ | $50.00_{\pm 0.00}$ | $45.00_{\pm 15.81}$ |
| Weighted Sampling | $45.00_{\pm 28.38}$ | $50.00_{\pm 0.00}$ | $45.00_{\pm 15.81}$ | $50.00_{\pm 0.00}$ | $100.00_{\pm 0.00}$ |

## 5.2 Experimental Results for Empirical Data Sets

Our experiments leverage the five empirical time series datasets on dynamic graphs from Qarkaxhija et al. (2022). This work also provides the optimal order of the higher-order model and the $\delta$ value (the maximum time difference for edges to be considered part of a causal walk) for generating the time respecting paths within each dataset. The data sets are *Highschool2011* and *Highschool2012* (Fournet & Barrat, 2014), *Hospital* (Vanhems et al., 2013), *StudentSMS* (Sapiezynski et al., 2019), and *Workplace2016* (Génois et al., 2015). These data sets are in particular relevant due to the following properties: They are

continuous-time data sets for static node classification and they include a sufficient large number of interactions compared to the number of nodes and edges. In our evaluation, we do not use the datasets from the TGB data sets (Huang et al., 2023) because they focus on time varying node labels. Our work does not address the prediction of changes of node labels in time, but the prediction a static node label based on sequential information. This section addresses the question of how our architecture compares to the described baselines with respect to the named empirical data sets. The mean balanced accuracy and its standard deviation is reported in Table 2.

We reproduce the superior results of DBGNN compared to other baselines for all data sets except Workplace2016 for which LGNN performs better than shown by Qarkaxhija et al. (2022). The obtained standard deviations are also comparable to the named work. However, HYPA-DBGNN outperforms all baselines, including DBGNN and TGN. For *Highschool2011* and *Highschool2012*, the gain is smallest with 2.77% and 2.27%, respectively. For *StudentSMS* and *Workplace2016*, the gain is about twice as large at 5.09% and 4.58%, respectively. It is noteworthy that the baseline results for *Workplace2016* are already at least 20% better than for the other data sets, so the gain of 4.58% is harder to achieve and brings the balanced accuracy close to the optimum. A remarkable result is the gain of 45.50% for *Hospital*. Here, the baselines are the weakest compared to the other data sets, while for our approach only *Workspace2016* is better solvable. Further, HYPA-DBGNN outperforms TGN in all tested cases. This difference can be explained by the observation that TGN does not use time-respecting paths that encode relevant patterns but rather accounts for the temporal evolution of the graph which is more relevant in the dynamic case. All in all, the inclusion of path anomalies is beneficial for all empirical data sets considered, but the gain depends on the particular data set. Here, the results for *Hospital* and *Workplace2016* stand out.

Table 2: Comparison of HYPA-DBGNN with node representation learning and deep graph learning baselines for dynamic graphs. The table presents the balanced accuracy and its standard deviation for the models on empirical static node classification tasks for dynamic graphs that is obtained through the outlined experiments. The best results are marked. Results with additional metrics are attached in Appendix J.

| Model | Highschool2011 | Highschool2012 | Hospital | StudentSMS | Workplace2016 |
|---|---|---|---|---|---|
| EVO | $43.68 \pm 10.91$ | $50.05 \pm 7.30$ | $25.83 \pm 8.29$ | $55.05 \pm 6.39$ | $26.50 \pm 12.08$ |
| HONEM | $59.00 \pm 10.61$ | $50.49 \pm 9.31$ | $39.44 \pm 17.57$ | $53.81 \pm 7.28$ | $83.17 \pm 11.14$ |
| DeepWalk | $54.64 \pm 17.70$ | $49.65 \pm 12.97$ | $24.58 \pm 10.92$ | $52.78 \pm 7.83$ | $20.54 \pm 9.51$ |
| Node2Vec | $54.64 \pm 17.70$ | $49.65 \pm 12.97$ | $24.58 \pm 10.92$ | $52.31 \pm 7.70$ | $20.54 \pm 9.51$ |
| GCN | $55.00 \pm 13.37$ | $59.35 \pm 11.13$ | $43.47 \pm 9.03$ | $54.50 \pm 6.40$ | $73.33 \pm 12.60$ |
| LGNN | $57.72 \pm 9.85$ | $51.43 \pm 17.94$ | $44.03 \pm 9.03$ | $52.71 \pm 6.63$ | $84.83 \pm 14.77$ |
| DBGNN | $61.54 \pm 11.13$ | $64.93 \pm 15.26$ | $52.50 \pm 19.27$ | $57.72 \pm 5.29$ | $84.42 \pm 15.59$ |
| TGN | $61.52 \pm 11.25$ | $41.52 \pm 6.19$ | $50.27 \pm 14.83$ | $50.67 \pm 4.10$ | $80.16 \pm 18.71$ |
| HYPA-DBGNN | $\mathbf{63.25} \pm \mathbf{16.18}$ | $\mathbf{66.41} \pm \mathbf{10.24}$ | $\mathbf{76.39} \pm \mathbf{17.12}$ | $\mathbf{60.66} \pm \mathbf{6.11}$ | $\mathbf{88.29} \pm \mathbf{10.51}$ |

### 5.3 Comparison of Temporal Sequences in Empirical and Synthetic Data

The synthetic data set encodes a pattern of increased class-assortativity that is learned by HYPA-DBGNN. Figure 2 shows the deviation from the expected edge frequencies in terms of HYPA scores for the used data sets regarding the incident nodes, i.e. for each node the distribution of the average HYPA score of incident edges is plotted. The second-order plot shows the increased class-assortativity for nodes of class 0 in the *Weighted Sampling* data set. The incident second-order edges have on average a larger HYPA score and thus are more often overrepresented compared to edges incident to nodes of class 1. Due to the statistic principled inferred graph, HYPA-DBGNN is able to learn this pattern. Also, *Hospital* and *Workplace2016* emit such under- and overrepresented sequential patterns in both graphs that are related to distinct node classes. In *Hospital* second-order edges incident to nodes of class 0 and 1 are overly often overrepresented. However, the first-order edges incident to nodes of class 0 and 1 differ in its statistics. This observed connection between node classes and the sequential patterns containing the respective nodes supports the superior performance of HYPA-DBGNN for *Hospital* and *Workplace2016*.

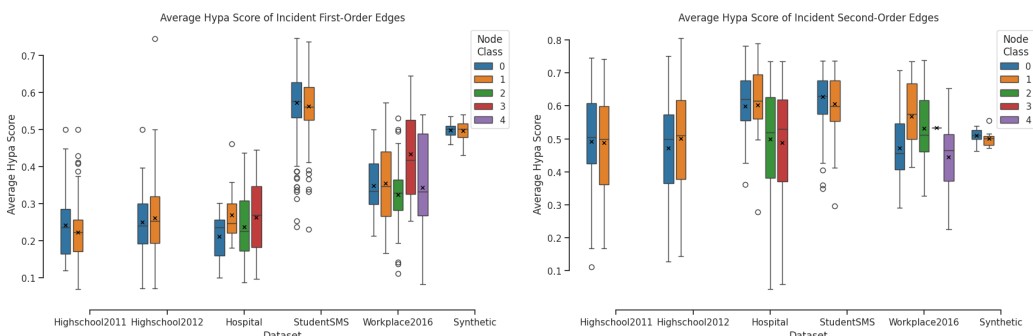

Figure 2: We plot the average HYPA score of all incident edges for each node and show the distribution with respect to nodes' classes. The left plot shows first-order HYPA scores and the right second-order ones. The synthetic set uses *Weighted Sampling*. For some data sets, varying distributions suggest a connection between the HYPA scores and the node classes.

## 6 CONCLUSION

In this work, we propose HYPA-DBGNN, a novel deep graph learning architecture that accounts for time-respecting paths in temporal graph data with high temporal resolution. Different from existing graph learning methods that employ neural message passing along time-respecting paths, we introduce a two-step approach which first infers anomalous sequential patterns based on an analytically tractable null model for time-respecting paths that preserves both the topology and the frequency, but not the temporal ordering, of time-stamped edges. In a second step, we apply neural message passing on an augmented higher-order De Bruijn graph, whose edges capture time-respecting paths that are overrepresented compared to the expectation from that random baseline. An experimental evaluation of our approach in a synthetic model and five empirical data sets on temporal graphs reveals that our proposed method considerably improves node classification compared to eight baseline methods in all studied data sets, with performance gains ranging from 2.27 % to 45.5 %. An investigation of HYPA scores – which capture the degree to which time-respecting path statistics deviate from what is expected in a null model – as well as an ablation study show that the correlation between node classes and the magnitude of the deviations from the random expectation is particularly pronounced for those empirical temporal graphs where we also observe the largest performance gains for our method. This finding highlights that the innovative combination of statistical inference and neural message passing, which is the key contribution of our work, leads to considerable advantages for temporal graph learning.

Despite these contributions, our work raises a number of open questions that we did not address within the scope of this work. First, in order to isolate the influence of sequential patterns in temporal graphs, here we solely focused on the sequence of time-stamped edges, thus neglecting additional node attributes and edge features. Future studies building on our work could thus additionally consider richer node and edge information, which is likely to further improve the performance of our model. Moreover, the framework of hypergeometric statistical ensembles allows to include non-homogeneous "edge propensities" based, e.g., on a homophily of nodes with similar attributes. This could possibly be used to generate domain-specific null models leading to a graph learning architecture that includes a non-trivial inductive bias, which we did not explore in this work. Bridging the gap between the application of statistical graph ensembles in network science and deep graph learning, we finally argue that our work opens broader perspectives for the integration of statistical graph inference, graph augmentation, and neural message passing. In particular, applying our method to the inference of (first-order) edges in static graphs could be a promising approach to address the issue that empirical graphs are rarely unspoiled reflections of reality, but are often subject to measurement errors and noise. The need to combine graph inference techniques with neural message passing (Ma et al., 2019; Pal et al., 2020; Zhang et al., 2019) has recently been identified as a major challenge for deep graph learning, and our work can be seen as a step in this direction.

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

## A    Variants of HYPA-DBGNN

In this section, we present other variations of our main HYPA-DBGNN architecture.

### A.1    Base Architecture without Anomalies (HYPA-DBGNN$^-$)

Replacing the HYPA scores with the absolute edge frequencies in the message passing procedure leads to the original message passing layers proposed by Kipf & Welling (2017). The overall structure including the bipartite layers is kept. The comparison of this model (HYPA-DBGNN$^-$) with HYPA-DBGNN reinforces the understanding of the significance of HYPA scores.

### A.2    Edge Embedded HYPA Scores (HYPA-DBGNN$^E$)

For HYPA-DBGNN the HYPA scores are used in a graph model selection step to enhance the message passing. Whereas for HYPA-DBGNN$^E$ the HYPA scores are understood as additional edge attributes whose significance is learned by an adapted graph convolution operation that embeds the edge attributes into the incident node attributes during message passing in the first graph neural network layers. The augmented propagation rule is given as

$$\vec{h}_{v_i}^{k,1} = \sigma \left( \sum_j \frac{1}{c_{ij}} \left( \vec{h}_{v_j}^{k,0} W^{k,1} + \vec{h}_{e_{ij}^k} W^{k,e} \right) \right), \tag{4}$$

with the first hidden representation $\vec{h}_{v_j}^{k,0}$ of node $u \in V^{(k)}$, the inferred HYPA scores in $\vec{h}_{e_{ij}^k}$ for the $k$-th-order edge $e_{ij}^k \in E^{(k)}$, the trainable weight matrices $W^{k,1} \in \mathbb{R}^{H^1 \times H^0}$ for the nodes and $W^{k,e} \in \mathbb{R}^{H^1 \times 1}$ for the edges and the normalization factor $c_{ij}$ as defined by Kipf & Welling (2017).

### A.3    Z-Score as Replacement for HYPA Scores (HYPA-DBGNN$^Z$)

The HYPA scores are based on the CDF. A a replacement for the CDF, a transformed Z-score instead of the HYPA score is implemented in HYPA-DBGNN$^Z$. The underlying soft configuration model provides the needed expected value and variance with

$$\mathbb{E}[X_{ij}] = m \frac{\Xi_{ij}}{M} \tag{5}$$

and

$$Var[X_{ij}] = m \frac{M-m}{M-1} \frac{\Xi_{ij}}{M} \tag{6}$$

needed to define the Z-score as

$$z(A_{ij}) = \frac{A_{ij} - \mathbb{E}[X_{ij}]}{\sqrt{Var[X_{ij}]}}. \tag{7}$$

Opposing to the HYPA score the Z-score is unbounded and possibly negative. Edges with negative Z-score are excluded because they are under-represented. Likewise in HYPA-DBGNN in most cases under-represented edges are removed, too, because their HYPA scores is approximately zero. Additionally, edges with a Z-score smaller than one are removed with the same argument of not having an unexpected large contribution to the graph and only beeing larger than 0 due to noisy fluctuations in the frequencies. The resulting restricted Z-score is logarithmically transformed due to observed large spread in empirical data, leading to the final replacement for the HYPA-score:

$$z'(e_{ij}) = \begin{cases} 0 & \text{if } z(e_{ij}) < 1, \\ \log(z(e_{ij})) & \text{otherwise} \end{cases} \tag{8}$$

## B   ABLATION STUDY - IMPACT OF STATISTICAL INFORMATION

We conduct an ablation study in which we compare our architectures HYPA-DBGNN, HYPA-DBGNN$^E$ and HYPA-DBGNN$^Z$ to the base architecture HYPA-DBGNN$^-$ that is not using statistical information. We aim to answer the question of what effect the addition of statistical information has on the prediction capability of the architectures in Table 3.

By comparing HYPA-DBGNN to HYPA-DBGNN$^-$ we see that the statistical information play an important role for all data sets but most importantly it becomes visible that the improvements for *Hospital* are indeed related to the additional information.

HYPA-DBGNN$^E$ with edge encoded statistical features performs better than the uninformed baseline but is most of the time significant weaker than HYPA-DBGNN. The structural graph correction applied in HYPA-DBGNN is still missing even when the edge encoder is able to learn the significance of the HYPA scores. HYPA-DBGNN$^Z$ performs weak for data sets where we don't see direct patterns in the analysis but works well for *Hospital*. It needs to be explored why the Z-score is more susceptible for data sets with weak or no patterns.

Table 3: Ablation study for HYPA-DBGNN. The best results are marked.

| Model | Highschool2011 | Highschool2012 | Hospital | StudentSMS | Workplace2016 |
|---|---|---|---|---|---|
| HYPA-DBGNN | **63.25** $_{\pm 16.18}$ | **66.41** $_{\pm 10.24}$ | **76.39** $_{\pm 17.12}$ | **60.66** $_{\pm 6.11}$ | 88.29 $_{\pm 10.51}$ |
| HYPA-DBGNN$^E$ | 61.54 $_{\pm 13.62}$ | 64.94 $_{\pm 17.71}$ | 59.03 $_{\pm 12.72}$ | 60.46 $_{\pm 9.42}$ | **88.50** $_{\pm 13.57}$ |
| HYPA-DBGNN$^Z$ | 53.97 $_{\pm 17.59}$ | 59.63 $_{\pm 15.74}$ | 69.31 $_{\pm 11.74}$ | 53.45 $_{\pm 7.50}$ | 88.42 $_{\pm 10.88}$ |
| HYPA-DBGNN$^-$ | 57.67 $_{\pm 17.16}$ | 64.49 $_{\pm 15.27}$ | 55.83 $_{\pm 19.27}$ | 56.23 $_{\pm 10.41}$ | 86.46 $_{\pm 12.65}$ |

## C   MODEL ARCHITECTURE

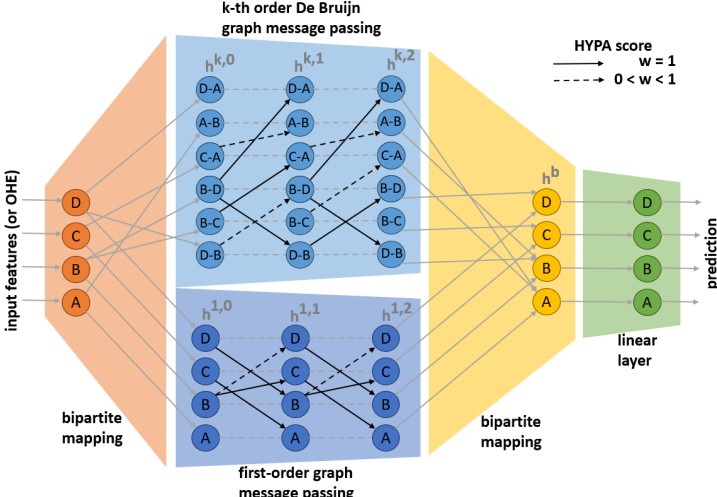

Figure 3: Illustration of the HYPA-DBGNN architecture. The architecture uses node features as inputs. In our case the node id is given as a one-hot encoding even though present features might be used. The bipartite mapping propagates the first-order node features to the first- and higher-order graph. The edge weights are given as HYPA scores such that the HYPA scores define the graph used for neural message passing. Under-represented edges with $HYPA^{(k)}(u,v) = 0$ are removed from the graph. Hence the used graphs are defined through the statistical model. The second bipartite layer merges the lower- and higher-order embedding after three message-passing layers. A final linear layer converts the embedding to the class prediction.

# D    Comments on Computational Complexity

There are two distinct steps to be considered when arguing about the complexity of our approach. First, there is the preprocessing step that creates the augmented graphs,i.e., the competition of the HYPA scores and the removal of the under-represented paths. Second, the graph neural network is trained on that graphs. For both steps, the complexity is determined by the number of edges in the higher-order De Bruijn graph. In the preprocessing, we calculate the HYPA score for higher-order edges.

The worst-case for the number of higher-order edges is given by the number of different sequences of length $k$, i.e., $|V|^k$ for a network with $|V|$ nodes. However, two arguments show that we can expect much lower complexity in real-world data. First of all, real-world networks are usually sparse, which implies that most sequences cannot occur as they would otherwise violate the network topology.

LaRock et al. (2020) use this argument, and prove that the complexity of their algorithm can be tightened with $\Delta G^{(k)} \leq |V|^2 \lambda_1^k$, where $|V|$ denotes the number of nodes in the first-order graph $G$ and $\lambda_1$ is the leading eigenvalue of the binary adjacency matrix of $G$. They conclude, that the HYPA score calculation scales linearly with the number of paths $N$ in the given data set for sparse real-world graphs, a moderate order $k$, and a sufficiently large $N$. (Qarkaxhija et al., 2022) also uses the argument of sparsity to further limit the complexity of the De Bruijn graph. They note that the number of walks of length $k$ becoming higher-order edges in the higher-order De Bruijn graph is also limited by $\sum_{ij} A_{ij}^k \leq |V|^k$, where $A^k$ is the k-th power of the binary adjacency matrix $A$ of $G$.

Furthermore, higher-order networks are even sparser than what we would expect based on the first-order topology. This is because the number of different time-respecting paths occurring on a network is generally much lower than the number of possible paths. (Qarkaxhija et al., 2022) demonstrate this (see in the appendix) by plotting the number of realized walks at each length and showing that in empirical graphs only a small fraction of walks is realized due to the restriction to time-respecting paths. By studying the complexity of the used empirical data set, they argue that De Bruijn graphs are applicable to real-world tasks.

We consider a path data set $S$ with $N$ entries. The number of edges in the k-th-order De Bruijn graph is denoted as $\Delta G^{(k)}$. LaRock et al. (2020) state that the asymptotic runtime of HYPA is $O(N + \Delta G^{(k)})$. A trivial upper-bound for $\Delta G^{(k)}$ is the fully connected case with $|V|^{k+1}$. This trivial case is also considered by Qarkaxhija et al. (2022) when they argue that the complexity of message passing on the De Bruijn graph is bounded.

# E    Experiment Resources and Reproducibility

We performed the experiments on a single PC with an NVIDIA GeForce RTX 3070 with 8 GB memory. On average one single experiment repetition takes approximately 5 minutes depending on the method and the data set. We run 4 experiments in parallel. We test the 12 methods (9 in the main study, 3 in the ablation study) with a parameter search over at most 25 variants on 7 data sets (5 empirical, 2 synthetic). All in all, the estimated time for the experiments is approximately $12 \cdot 7 \cdot 25 \cdot 5/60 \approx 440$ hours, excluding pre-studies. While this is only a rough estimate it reflects the order of magnitude of time needed to run all experiments.

To reproduce the experiments, we provide a reference implementation at [blinded] together with synthetic and empirical data sets and their splits and licenses. For the implementations of the baselines we attribute the reused implementations from the DBGNN reference paper (Qarkaxhija et al., 2022). They also parse and provide the used empirical data sets.

We include a self-containing benchmark to compare HYPA-DBGNN to other methods including strong candidates like GCN and DBGNN following the described evaluation procedure. The benchmark is as concise as possible to let the reader focus on the main contributions. This benchmark can be used to reproduce presented results.

## F    Properties of Experiment Data Sets

Table 4: Overview of time series data and ground truth node classes used in the experiments. $\delta$ describes the maximum time difference for edges to be considered part of a casual walk.

| Data Set | Ref. | Events | $|V|$ | $|E|$ | $|V^{(2)}|$ | $|E^{(2)}|$ | Classes (Sizes) | $\delta$ |
|---|---|---|---|---|---|---|---|---|
| Highschool2011 | (Fournet & Barrat, 2014) | 28561 | 126 | 3355 | 3042 | 17141 | 2 (85/41) | 4 |
| Highschool2012 | (Fournet & Barrat, 2014) | 45047 | 180 | 4399 | 3965 | 20614 | 2 (132/48) | 4 |
| Hospital | (Vanhems et al., 2013) | 32424 | 75 | 2052 | 2028 | 15500 | 4 (29/27/11/8) | 4 |
| StudentSMS | (Sapiezynski et al., 2019) | 24333 | 429 | 1160 | 733 | 846 | 2 (314/115) | 40 |
| Workplace2016 | (Génois et al., 2015) | 9827 | 92 | 1491 | 1431 | 7121 | 5 (34/26/15/13/4) | 4 |
| Random Sampling | Ours (Synthetic) | 8388608 | 20 | 400 | 400 | 1600 | 2 (10/10) | 1 |
| Weighted Sampling | Ours (Synthetic) | 8388608 | 20 | 400 | 400 | 1600 | 2 (10/10) | 1 |

## G    TGN Adaptations

We implement TGN as proposed by Rossi et al. (2020) Instead of a link prediction layer, we add a node prediction layer as the last stage. The embedding size is fixed to 32 as for the other models. For TGN the training procedure is adapted due to its dynamic origin. The proposed training procedure for dynamic node predictions splits the events into fixed-size temporal batches and predicts the next node state for the nodes affected by the events. The batches are temporally divided into train and test batches. Opposing, the static prediction task splits the nodes into train and test sets. We try to keep as much from the original training procedure as possible to favor the memory based architecture. Hence, we train the model on all event batches of size 200 but restrict the training nodes to the train set with fixed class. The last prediction for the given test nodes is used to evaluate the performance. This is not necessary in the last batch of events. For the synthetic data the batch size is increased to 200.000 since each of the $2^{23}$ events has its own timestamps which leads to infeasible training time with lower batch sizes. Compared to the other deep learning methods the model the losses are updated more often because they are updated for every event batch and not only for every node batch. Consequently, we adapt the learning rate to 0.0001 and the originally used optimizer Adam to obtain improved results.

## H    Ablation Study - Impact of Individual Parts

Table 5: Ablation study for HYPA-DBGNN showing the balanced accuracy. Subsequently parts of the model are removed. (a) contains the complete HYPA-DBGNN model. In (b) the HYPA scores are removed such that the statistical information are not passed to the model. In (c) we further remove the first bipartite layer that maps the first-order node features to the second-order nodes and replace it by a second-order one-hot encoding (OHE). In (d) we additionally remove the complete second-order message passing (MP).
In (e) we use the base HYPA-DBGNN but replace the first-order OHE with available features. Only Highschool2011 and Highschool2012 contain node features. Those are the classes the students belong to. We suspect that those features are not informative for the given prediction task.

| Model | Highschool2011 | Highschool2012 | Hospital | StudentSMS | Workplace2016 |
|---|---|---|---|---|---|
| (a) base HYPA-DBGNN | $63.25 \pm 16.18$ | $66.41 \pm 10.24$ | $76.39 \pm 17.12$ | $60.66 \pm 6.11$ | $88.29 \pm 10.51$ |
| (b) without HYPA scores | $57.67 \pm 17.16$ | $64.49 \pm 15.27$ | $55.83 \pm 19.27$ | $56.23 \pm 10.41$ | $86.46 \pm 12.65$ |
| (c) OHE instead bipartite layer | $61.54 \pm 11.13$ | $64.93 \pm 15.26$ | $52.50 \pm 19.27$ | $57.72 \pm 5.29$ | $84.42 \pm 15.59$ |
| (d) without second-order MP | $55.00 \pm 13.37$ | $59.35 \pm 11.13$ | $43.47 \pm 9.03$ | $54.50 \pm 6.40$ | $73.33 \pm 12.60$ |
| (e) HYPA-DBGNN with features | $59.12 \pm 20.24$ | $62.43 \pm 10.06$ | - | - | - |

## I    Synthetic Data Creation Procedure

In this section, we give information about the synthetic data set creation and its characteristics We use two synthetic data sets that are created with the following procedure. Figure 4 gives an overview of the procedure.

The algorithm consists of two main parts aimed at constructing the first-order and second-order topology of the network, respectively. Initially, the algorithm receives as input parameters the set of nodes, a node-to-class mapping, a bias parameter, and the desired number of paths of length $k$ (k-th order edges) to generate.

In the first part, we assign the node degrees, and consequently the values of the $\Xi$ matrix as $\Xi = k_{in} \cdot k_{out}$, To do this, we give each node a random weight sampled from a continuous uniform distribution $\mathcal{U}[0,1]$. Next, for each node, we sample a number of (unweighted) edge stubs from a multinomial distribution. The number of categories in the multinomial distribution equals the number of nodes, and the probability for each category, respectively edge stub, is proportional to the previously assigned node weight. The number of stubs we sample equals the desired number of paths of length $k$ given in input. Once we have this, we randomly connect the in and out stubs, thus getting the multi-set of multi-edges and the first-order topolgy. Notice that the multi-edges created in this step also yields the higher-order nodes, and that the multi-edge frequencies correspond to their in- and out-weighted degrees.

In the second part, an iterative process creates higher-order edges. First, an out-stub $(\langle v_0 v_1 \ldots v_{k-1} \rangle, \cdot)$ is sampled proportional to its weighted out-degree. Subsequently, a set P of potential in-stubs $(\cdot, \langle v_1 \ldots v_{k-1} v_k \rangle)$ is identified, ensuring valid connections between higher-order nodes by applying the de Bruijn condition that requires the last $k-1$ elements of $(\langle v_0 v_1 \ldots v_{k-1} \rangle, \cdot)$ to match the first $k-1$ elements of $(\cdot, \langle v_1 \ldots v_{k-1} v_k \rangle)$. The sampling process for successor in-stubs from P is biased based on the classes of the first-order nodes $v_0$, $v_1 \ldots v_{k-1}$, and $v_k$. Specifically, counts are artificially inflated by the bias parameter for in-stubs where all $k$ nodes belong to the same class, encoding the desired pattern of preferential attachment. The selected out-stub $(\langle v_0 v_1 \ldots v_{k-1} \rangle, \cdot)$ and in-stub $(\cdot, \langle v_1 \ldots v_{k-1} v_k \rangle)$ form a higher-order edge $(\langle v_0 v_1 \ldots v_{k-1} \rangle, \langle v_1 \ldots v_{k-1} v_k \rangle)$ in the final network. This iterative process continues until all stubs are connected, resulting in paths of length $k$ that predominantly connect nodes within the same class, with the degree of class-assortativity controlled by the bias parameter.

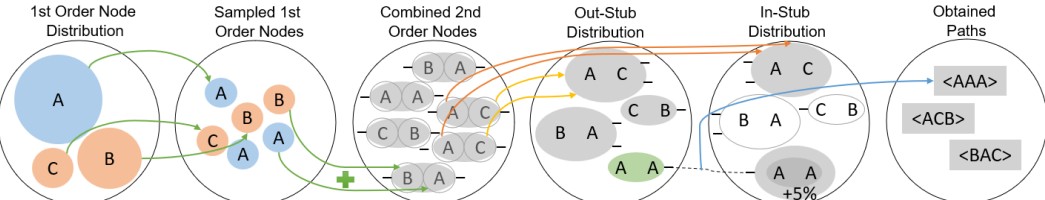

Figure 4: This figure presents the sampling procedure for the synthetic path data. It consists of five steps (left to right): (1) *sampling* of first-order nodes (uniform distribution) from a set with two classes (blue and orange); (2) *combining* the sampled nodes into second order nodes; (3) *sampling out-connection* candidates from the set of second-order nodes (e.g., $(\langle A, A \rangle, \cdot)$ highlighted in green). (4) *sampling in-connections* for every out-stub we sample a valid in-stub (e.g., from $(\langle A, A \rangle, \cdot)$: $(\cdot, \langle A, C \rangle)$ or $(\cdot, \langle A, A \rangle)$ – highlighted in grey). Valid in-stubs whose nodes belong to the same group have a 5% increased probability of being sampled $((\cdot, \langle A, A \rangle)$ gets the bonus while $(\cdot, \langle A, C \rangle)$ does not). (5) the edges are saved as paths $(\langle A, A, A \rangle)$.

## I.1 Synthetic Data Characteristics

We use two synthetic data sets with $n = 2^{22}$ paths. The paths emit first and second-order graphs with heterogeneous edge statistics. Figure 5 presents the the edge statistics for the synthetic data sets. For the given resolution, the graph for the synthetic data set with implanted pattern looks identical to the one without pattern due to the construction through the reused expected first-order statistics that defines the $\Xi$-matrix for the second-order statistics and a sufficient small bias parameter during sampling. However, the emitted edge frequencies vary between the emitted graphs due to the random sampling procedure.

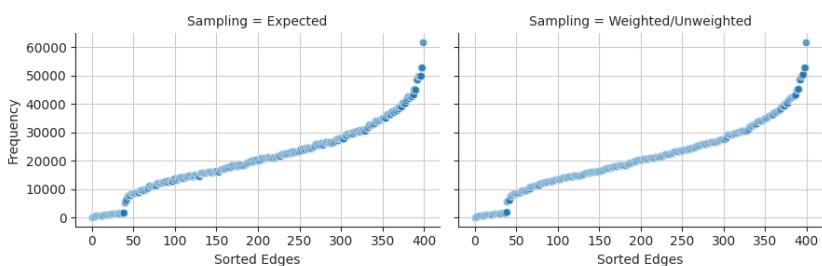

(a) First-order edge statistics. They are equal for both the *Weighted* and the *Unweighted* data set. The differences are not observable by comparing them with the expected first-order edge statistics. The heterogeneous distribution is clearly visible.

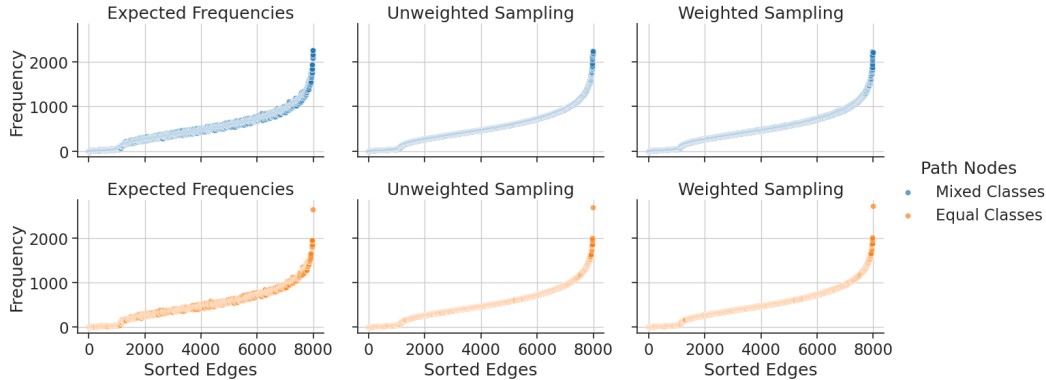

(b) Second-order edge statistics. The frequencies in the *Weighted* and *Unweighted* data sets differ but it is not visible due to the heterogeneous distribution. They also differ from the expected frequencies. We also distinguish between paths connecting same class nodes and different class nodes. Here it becomes clear that the mere frequencies – that are skewed – are not enough to distinguish between both cases.

Figure 5: Edge frequencies of the emitted graphs for the synthetic data sets. The plots show that due to the heterogeneous distribution overrepresented paths do not become visible. Figure 6 gives a zoomed in view to show the differences exploited by HYPA-DBGNN.

Figure 6 presents the absolute difference of the first- and second-order edge frequencies between the two data sets. Notable, all edges whose incident nodes are predominantly connected are sampled more often due to the bias parameter. This class-assortativity needs to be learned by the machine learning model.

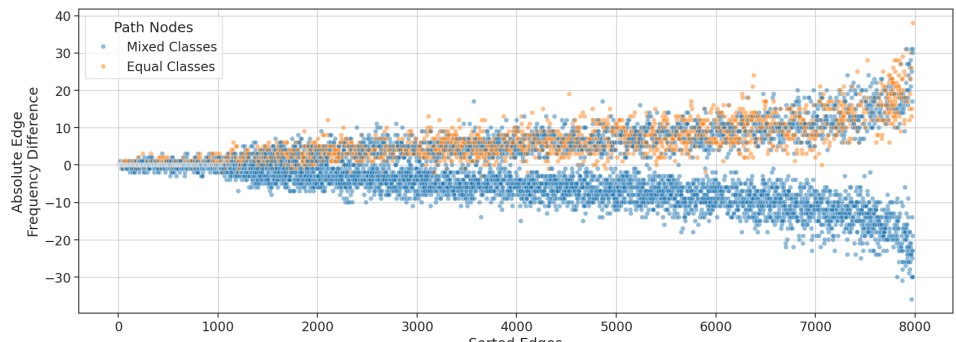

Figure 6: This plot presents the absolute difference of the second-order edge frequencies of the *Weighted* and *Unweighted* data set. Due to the random sampling there are edges that have a higher frequency in on or the other data set. This trend increases with for edges that have more candidates in the urn. The edges that represent paths connecting nodes from the same class are mostly more often sampled and thus overrepresented in the *Weighted* data set. However, compared to the absolute frequencies in Figure 5 the deviations are minor such that edges with low frequencies can be overrepresented. HYPA-DBGNN learns this pattern.

The comparison in Figure 7 of the frequencies with the the expected frequencies given by the Ξ-matrix supports the differences between the two synthetic data sets and highlights the encoded class-assortativity in the data set with the biased sampling.

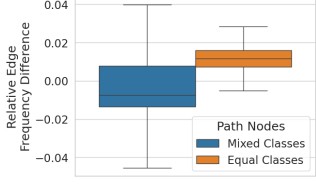 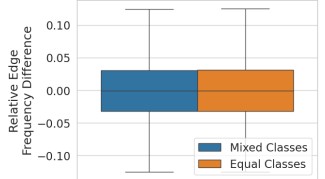 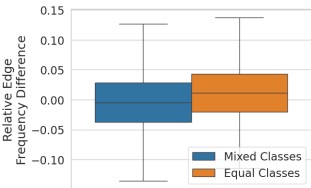

(a) Relative frequency difference of the same second-order paths between the *Unweighted Sampling* and *Weighted Sampling* data. Paths connecting same class nodes on average have a higher frequency in the *Weighted Sampling* data set. This is consistent with Figure 6.

(b) Relative frequency difference of the same second-order paths between the *Unweighted Sampling* data and the expected path frequencies. Here, no bias parameter is applied. Thus, the frequencies of paths connecting same class nodes are vary as much from the expected frequencies than the other paths.

(c) Relative frequency difference of the same second-order paths between the *Weighted Sampling* data and the expected path frequencies. An increased bias parameter is applied. Thus, the frequencies of paths connecting the same class nodes appear more often with respect to the expected frequencies than the other paths.

Figure 7: Box plots showing how the distribution of second-order path frequencies vary in comparison between the two synthetic data sets and in comparison to the expected path frequencies. Only in the *Weighted Sampling* data set, the paths connecting same class nodes appear more often than the remaining paths.

## J    ADDITIONAL RESULTS

Table 6: Comparison of our architectures (HYPA-DBGNN, HYPA-DBGNN$^-$, HYPA-DBGNN$^E$, HYPA-DBGNN$^Z$) with different machine learning models. The balanced accuracy is given in Table 1, Table 2 and Table 3. The results are obtained as described in Section 5. The best results are marked.

| Data Set | Model | F1-score-macro | Precision-macro | Recall-macro |
|---|---|---|---|---|
| Highschool2011 | EVO | $39.51 \pm 11.50$ | $39.38 \pm 19.64$ | $43.68 \pm 10.91$ |
| | HONEM | $57.54 \pm 11.52$ | $58.19 \pm 13.09$ | $59.00 \pm 10.61$ |
| | DeepWalk | $53.70 \pm 18.55$ | $53.47 \pm 19.61$ | $54.64 \pm 17.70$ |
| | Node2Vec | $53.70 \pm 18.55$ | $53.47 \pm 19.61$ | $54.64 \pm 17.70$ |
| | GCN | $48.55 \pm 15.49$ | $49.45 \pm 18.52$ | $55.00 \pm 13.37$ |
| | LGNN | $52.66 \pm 14.71$ | $53.57 \pm 15.97$ | $57.72 \pm 9.85$ |
| | DBGNN | $57.08 \pm 11.35$ | $61.78 \pm 10.75$ | $61.54 \pm 11.13$ |
| | TGN | $57.32 \pm 9.84$ | $59.56 \pm 10.59$ | $61.52 \pm 11.25$ |
| | HYPA-DBGNN | $\mathbf{59.60} \pm \mathbf{15.04}$ | $62.55 \pm 14.38$ | $\mathbf{63.25} \pm \mathbf{16.18}$ |
| | HYPA-DBGNN$^-$ | $55.92 \pm 17.41$ | $56.85 \pm 16.26$ | $57.67 \pm 17.16$ |
| | HYPA-DBGNN$^E$ | $57.30 \pm 15.77$ | $\mathbf{63.29} \pm \mathbf{14.85}$ | $61.54 \pm 13.62$ |
| | HYPA-DBGNN$^Z$ | $49.63 \pm 17.56$ | $52.23 \pm 19.82$ | $53.97 \pm 17.59$ |
| Highschool2012 | EVO | $46.83 \pm 9.44$ | $47.97 \pm 18.15$ | $50.05 \pm 7.30$ |
| | HONEM | $50.58 \pm 9.49$ | $53.89 \pm 15.27$ | $50.49 \pm 9.31$ |
| | DeepWalk | $48.79 \pm 13.02$ | $49.75 \pm 13.77$ | $49.65 \pm 12.97$ |
| | Node2Vec | $48.79 \pm 13.02$ | $49.75 \pm 13.77$ | $49.65 \pm 12.97$ |
| | GCN | $54.53 \pm 10.82$ | $56.94 \pm 12.00$ | $59.35 \pm 11.13$ |
| | LGNN | $45.32 \pm 16.88$ | $51.43 \pm 14.63$ | $51.43 \pm 17.94$ |
| | DBGNN | $60.22 \pm 13.73$ | $63.18 \pm 12.57$ | $64.93 \pm 15.26$ |
| | TGN | $38.32 \pm 5.37$ | $35.86 \pm 5.36$ | $41.52 \pm 6.19$ |
| | HYPA-DBGNN | $60.58 \pm 12.12$ | $\mathbf{66.23} \pm \mathbf{13.01}$ | $\mathbf{66.41} \pm \mathbf{10.24}$ |
| | HYPA-DBGNN$^-$ | $61.26 \pm 16.13$ | $64.37 \pm 15.44$ | $64.49 \pm 15.27$ |
| | HYPA-DBGNN$^E$ | $\mathbf{61.53} \pm \mathbf{17.30}$ | $64.22 \pm 15.56$ | $64.94 \pm 17.71$ |
| | HYPA-DBGNN$^Z$ | $56.00 \pm 15.24$ | $58.46 \pm 14.32$ | $59.63 \pm 15.74$ |
| Hospital | EVO | $20.05 \pm 6.64$ | $19.12 \pm 9.20$ | $25.00 \pm 7.86$ |
| | HONEM | $34.88 \pm 18.22$ | $36.88 \pm 23.53$ | $37.50 \pm 17.35$ |
| | DeepWalk | $20.00 \pm 9.53$ | $18.76 \pm 9.68$ | $23.89 \pm 10.91$ |
| | Node2Vec | $20.00 \pm 9.53$ | $18.76 \pm 9.68$ | $23.89 \pm 10.91$ |
| | GCN | $37.38 \pm 8.67$ | $33.83 \pm 8.00$ | $43.47 \pm 9.03$ |
| | LGNN | $35.81 \pm 8.96$ | $32.75 \pm 10.64$ | $44.03 \pm 9.03$ |
| | DBGNN | $47.87 \pm 20.02$ | $48.21 \pm 21.79$ | $51.67 \pm 20.34$ |
| | TGN | $46.50 \pm 13.60$ | $50.83 \pm 8.89$ | $49.16 \pm 16.95$ |
| | HYPA-DBGNN | $\mathbf{71.80} \pm \mathbf{19.18}$ | $\mathbf{71.50} \pm \mathbf{20.95}$ | $\mathbf{74.31} \pm \mathbf{17.45}$ |
| | HYPA-DBGNN$^-$ | $51.91 \pm 20.77$ | $50.83 \pm 22.33$ | $55.00 \pm 20.49$ |
| | HYPA-DBGNN$^E$ | $52.08 \pm 13.10$ | $52.25 \pm 13.41$ | $59.03 \pm 12.72$ |
| | HYPA-DBGNN$^Z$ | $65.66 \pm 13.39$ | $66.79 \pm 16.20$ | $69.31 \pm 11.74$ |
| StudentSMS | EVO | $54.62 \pm 7.73$ | $55.63 \pm 9.53$ | $55.05 \pm 6.39$ |
| | HONEM | $52.46 \pm 9.71$ | $55.65 \pm 14.29$ | $53.81 \pm 7.28$ |
| | DeepWalk | $52.08 \pm 7.19$ | $53.18 \pm 7.61$ | $52.78 \pm 7.83$ |
| | Node2Vec | $51.87 \pm 7.39$ | $52.13 \pm 6.90$ | $52.31 \pm 7.70$ |
| | GCN | $53.85 \pm 6.39$ | $54.39 \pm 6.27$ | $54.50 \pm 6.40$ |
| | LGNN | $46.79 \pm 5.27$ | $52.70 \pm 6.07$ | $52.71 \pm 6.63$ |
| | DBGNN | $56.87 \pm 5.05$ | $58.55 \pm 5.58$ | $57.72 \pm 5.29$ |
| | TGN | $48.98 \pm 4.50$ | $50.71 \pm 3.10$ | $50.67 \pm 4.10$ |
| | HYPA-DBGNN | $\mathbf{60.47} \pm \mathbf{6.68}$ | $\mathbf{61.40} \pm \mathbf{7.00}$ | $\mathbf{60.66} \pm \mathbf{6.11}$ |
| | HYPA-DBGNN$^-$ | $54.58 \pm 9.12$ | $55.66 \pm 8.88$ | $56.23 \pm 10.41$ |
| | HYPA-DBGNN$^E$ | $59.31 \pm 9.08$ | $59.97 \pm 9.24$ | $60.46 \pm 9.42$ |
| | HYPA-DBGNN$^Z$ | $52.60 \pm 6.74$ | $54.24 \pm 9.03$ | $53.45 \pm 7.50$ |
| Workplace2016 | EVO | $22.74 \pm 12.34$ | $21.84 \pm 14.18$ | $26.50 \pm 12.08$ |
| | HONEM | $77.75 \pm 11.70$ | $79.53 \pm 13.50$ | $79.46 \pm 10.32$ |
| | DeepWalk | $17.23 \pm 8.77$ | $16.30 \pm 9.42$ | $20.54 \pm 9.51$ |
| | Node2Vec | $17.23 \pm 8.77$ | $16.30 \pm 9.42$ | $20.54 \pm 9.51$ |
| | GCN | $68.56 \pm 14.78$ | $66.21 \pm 16.88$ | $73.33 \pm 12.60$ |
| | LGNN | $82.96 \pm 15.65$ | $84.32 \pm 15.04$ | $84.83 \pm 14.77$ |
| | DBGNN | $81.16 \pm 19.16$ | $81.33 \pm 20.14$ | $84.42 \pm 15.59$ |
| | TGN | $78.71 \pm 20.32$ | $79.95 \pm 22.21$ | $80.16 \pm 18.71$ |
| | HYPA-DBGNN | $85.82 \pm 12.23$ | $85.42 \pm 13.75$ | $88.29 \pm 10.51$ |
| | HYPA-DBGNN$^-$ | $82.75 \pm 14.26$ | $83.25 \pm 15.21$ | $84.71 \pm 13.66$ |
| | HYPA-DBGNN$^E$ | $86.47 \pm 16.28$ | $86.00 \pm 17.36$ | $\mathbf{88.50} \pm \mathbf{13.57}$ |
| | HYPA-DBGNN$^Z$ | $\mathbf{87.67} \pm \mathbf{11.99}$ | $\mathbf{88.83} \pm \mathbf{13.10}$ | $88.42 \pm 10.88$ |

