# OpenReview forum: "Inference of Sequential Patterns for Neural Message Passing in Temporal Graphs"
_ICLR.cc/2025/Conference — Submitted to ICLR 2025_

### Official Review · Reviewer_Y5VP · 2024-10-31

**Soundness:** 3
**Presentation:** 2
**Contribution:** 1
**Rating:** 5
**Confidence:** 4

**Summary:**

This paper introduces HYPA-DBGNN, a graph augmentation architecture focused on temporal graph learning. It encodes sequential pattern dynamics in first- and higher-order De Bruijn graphs and corrects graph structures using anomaly statistics. HYPA-DBGNN computes HYPA scores via hypergeometric ensembles to assess edge frequency differences from a random model, adjusting weights to improve accuracy. It uses a multi-order message passing scheme with inductive bias, incorporating HYPA scores and ReLU activation while preserving graph sparsity to optimize efficiency.

**Strengths:**

1. The paper introduces De Bruijn graphs into temporal graph analysis, which I find to be a novel approach.

2. The paper conducts extensive experiments to demonstrate the effectiveness of the proposed method.

**Weaknesses:**

1.The paper's exposition is not very clear, with many key pieces of information relegated to the appendices.

2.The paper does not clearly explain why the introduction of De Bruijn graphs enhances performance, making it seem more like a simple combination of existing methods.

3.The explanation of the method is insufficiently clear; a framework diagram could be helpful.

**Questions:**

1. Could the authors explain the role of De Bruijn graphs?

---

> ### Author Response · Authors · 2024-11-20
>
> We thank you for your helpful review and the positive comments about our work. We have addressed the main points in the aggregate response to all reviewers.
>
> To answer the question about the role of the De Bruijn graphs we like to refer to the framework diagram in appendix C. For clarity, we consider moving this to the main part of the manuscript. The higher-order De Bruijn graph encodes the transitive dependencies found in the data set. We enhance that information based on the described null model to make the model able to express even more patterns. Different orders of the De Bruijn graph are used to encode first- and higher-order dependencies.

---

> > ### Comment · Reviewer_Y5VP · 2024-11-26
> > **Thanks**
> >
> > Thank you to the authors for their responses. I will maintain my current rating.

---

### Official Review · Reviewer_UJFa · 2024-11-02

**Soundness:** 2
**Presentation:** 2
**Contribution:** 3
**Rating:** 5
**Confidence:** 3

**Summary:**

This work introduces a model termed HYPA-DBGNN, which seeks to improve the ability of a GNN in temporal settings to learn high-order time dependent interactions. HYPA-DBGNN has two components, HYPA which detects the ``surprise'' of observing a specific walk, and DBGNN which performs a hypergeometric walk feature extraction. The authors detail this model as an extension of DBGNN, and present experiments which show promising performance gains.

**Strengths:**

1. The paper is well organized and motivated.
2. The problem of extracting complex relationships from transitions between vertices is an interesting problem with many industrial applications
3. The experiments that are presented appear to be carefully performed and well motivated. The results as presented provide evidence that the method works.

**Weaknesses:**

1. The paper is unclear in spots. For example, The concept of a De Bruijn graph is mentioned but its basic properties are not discussed.
2. The mathematical notation is intricate and can be difficult to follow, with some symbols overlapping with standard symbols from the literature. For example, $H(v)$ is the sum of $HYPA$ factors but is traditionally the hidden representation for all vertices.
3. The intuition for
4. Minor typos and grammatical issues make the paper somewhat difficult to follow. For example, `fist` -> `first` on line 314.
5. Experiments in section 5.2 seem to lack many modern baselines including CAWN, TGAT, DySAT, and others. I would recommend that the authors add additional baselines. Random walk GNNs such as RWGNN could be applicable here as well, as could transformer architectures.
6. The experimental setup is unclear in spots, the baselines may have been untuned, and the graphs are small.

**Questions:**

1. Does this new inductive bias lead to a provably more expressive GNN than previous temporal MPNNs?
2. What is the run-time scaling of HYPA-DBGNN? All experiments were run on quite small graphs, so it's hard to understand how scalable of a technique this is.
3. To what extent has hyperparameter tuning been performed?
4. What explains HONEM's good performance in 5.1?

---

> ### Author Response · Authors · 2024-11-20
>
> We thank you for your insightful review and the positive comments about our work. We have addressed most of your questions in the aggregate response to all reviewers.
>
> .
>
> Regarding the **hyper parameters**, we like to refer to section 5.1 for an explanation of the used hyper parameters. As stated, we also tune the hyper parameters for the baselines. The same setup is used for tuning and evaluating our model and the baselines. In response to reviewer LQXN, we will move this explanation to the appendix and add another table to make the hyper parameter ranges and search even more clear.
>
> .
>
> Regarding the **choice of data sets** used in our evaluation:  this was based  on the need to have a sufficiently large number of observed interactions compared to the number of nodes and edges. This is necessary in order to observe a number of time-respecting paths that is sufficiently large to establish significant deviations from the expected values calculated from the model. Most available large data sets on temporal graphs have large numbers of nodes and edges, but are too sparse in terms of observed time-respecting paths.
>
> .
>
> Regarding the **performance of HONEM**:
>
> The synthetic data sets contain random fluctuations in the possibly higher-order edge frequency statistics due to the randomized creation and splitting. HONEM that heavily utilized higher-order edge frequencies seems to learn these fluctuations and not the pattern because the standard deviation is high and the performance does not increase for the data set with pattern.
> Also for the empirical data HONEM relies on the higher-order edge frequencies. As a result it performs better than first-order methods, especially in  data sets like Hospital or Workplace where the higher-order statistics are linked to the classification task (see Fig. 2).

---

> > ### Comment · Reviewer_UJFa · 2024-11-27
> >
> > Thank you for addressing my questions, but I didn't see any serious effort made to address any of the weaknesses that were pointed out. Because of this, I will be holding my score.

---

### Official Review · Reviewer_LQXN · 2024-11-04

**Soundness:** 2
**Presentation:** 1
**Contribution:** 1
**Rating:** 3
**Confidence:** 3

**Summary:**

This work focuses on relatively novel task, static node property classification for temporal graphs. Different from common trend of temporal graph neural networks, it proposes HYPA-DBGNN that extends a previous work GBGNN (which combines static hyper-order graph neural network on a high-order De Bruijn Graph constructed from time series) by null model correction.

**Strengths:**

- This work focuses on static node property classification on temporal graph, which is task lack of exploring.

**Weaknesses:**

- The notation is lack of consistency, making it hard to follow and the clarify of method details being quite poor. (See questions)
- This paper focuses on a rare task, which I think more real-world justification is needed? For example, what are real-word scenarios? You can pick one of your dataset to explain this in more detail.
- The contribution of the proposal is slightly unclear. It seems that this work simply extends related work DBGNN by introducing null model correction. If my understanding is correct, I think more theoretical justification of the necessity of this correction should be provided, otherwise, the contribution seems to be limited.
- In both the synthetic and real-world experiments, the variance are very large, makes me doubt if the problem is formalized correctly.
- Compare to highly related baseline DBGNN, the experiment results is not quite impressive (confident interval overlaps a lot in many tasks). This makes the contribution of null model correction less sound given no theoretical justification of the necessity.

**Questions:**

- Figure 1: Why we construct a higher-order edge for count 0? Besides, shouldn't we have an arrow from (a) to (d) since we also need 1-order counts to construct 1-order graph in (d)?
- Figure 1: You should extend the figure with how null model in (b) and weights in (c) are really generated, or provide in appendix? This figure fails to explain what you did for (b) and (c) given the poor explanation of section 4.
- line 262: What is $X_{uv}$ and $f(u, v)$? Why they are independent from order $k$?
- line 282: Shouldn't $H(v)$ rely on order $k$ based on your definition? Same to equation (1).
- Page 6: Mix use of higher order nodes and nodes make the notation is bit hard to follow, recommend to replace $v$ by $v^{(k)}$ in all related content, or vector form $\mathbf{v}$. Then, you can claim that $k = 1$ is omitted by default.
- line 292: Why map $h^{1, 0}$ to $h^{k, 1}$ rather than $h^{k, 0}$?
- line 295: Can you provide more explanation how this bipartition is analogous to Markov chain?
- line 304: What is $g$?
- Why this design is limited to temporal node classification? I think this architecture can be used for regression without any modification.
- line 331-351: Hyperparamter configuration can be moved to appendix so that you can have more space to improve clarity of algorithm design sections.
- Experiment: You are comparing with a lot of simple baselines for static graph with only on temporal graph baseline. Based on [1], static and temporal graph representation are indeed equivalent, especially you are performing static node classification on temporal graph. Why you don't compare with other basics such as GAT, GIN, TGAT, DySAT (see [1]), and other state-of-the-art like PNA, PINE, GraphTransformer.
- Given that TGN is designed mainly for evolving graph, should you make some modification to make comparison fair? For example, average node representation of different timestamps for perform static node classification on temporal graph?
- Your font looks different from template. I think you need to check if you are using the template correctly.

[1] Gao, Jianfei, and Bruno Ribeiro. "On the equivalence between temporal and static equivariant graph representations." International Conference on Machine Learning. PMLR, 2022.

---

> ### Author Response · Authors · 2024-11-20
>
> We thank you for the detailed review and the check of our notation. In the following we will focus on the questions except for the notation. But we like to highlight that those notation questions substantially improve our manuscript. The remaining questions are answered in the general response.
>
> Regarding your questions:
>
> .
>
> **For example, what are real-word scenarios**?
>
> Our work exploits patterns in dependencies, e.g. in interactions of social networks. Those are relevant in different domains.
>
> In the following we refer to the workplace and hospital data sets. Here, we predict the role of the employees. With this prediction, we can make suggestions for new positions in the company based on the interaction patterns. This knowledge gained from interaction patterns can also be used to retrain employees to better fit their position. Both measures may increase the efficiency of the company.
>
> Another not yet mentioned example is the detection of fraud in a financial context. Companies like [Robinhood  Europe, U.A.B.](https://newsroom.aboutrobinhood.com/preventing-fraud-at-robinhood-using-graph-intelligence/?utm_source=blog.quastor.org&utm_medium=referral&utm_campaign=how-robinhood-uses-graph-algorithms-to-prevent-fraud) monitor trading movements and want to identify malicious members. Here a null model based approach can be essential to give important insights. Even though we are not aware of public benchmark data sets in this area we consider this as a very important use case that is not deeply explored, yet.
>
> We thank you for this question and improve the motivation in the final manuscript.
>
> .
>
> **Can you provide more explanation how this bipartition is analogous to Markov chain**?
>
> The higher-order graph models dependencies over multiple steps. The first bipartite layer maps the first-order dependency to the corresponding higher-order dependency that continues the chain.  The final bipartite layer maps the higher-order dependency to the corresponding first-order dependency that again continues the chain. Hence, no information from the dependencies more than one step away are passed to the succeeding dependencies.
>
> .
>
> **Why this design is limited to temporal node classification? I think this architecture can be used for regression without any modification.**
>
> We agree that there is no technical limitation to restrict or model to node classification. Static regression tasks for graphs with temporal patterns are an interesting avenue. However, to this point, we are not aware of suitable benchmark data sets. We are delighted to discuss further directions and references in this area.
>
> .
>
> All in all, we like to thank you for the very detailed review and suggestions that result in a lasting improvement to our manuscript.

---

> > ### Comment · Reviewer_LQXN · 2024-11-23
> >
> > I acknowledged your reply, and have briefly reviewed other response.
> > I understand the limitation of your experiments, and I can hold my points of adding new baselines and datasets (while I still believe it is necessary to improve paper quality).
> > However, I still need several points to be addressed to improve my score from high to low:
> > 1. Based on your response, I understand the task is improving static node prediction on temporal graph by a method with better (potential) explainability. However, I don't think the experiment is sound enough to claim this: As I said, **your experimental results on real datasets has giant confidence interval, and mix with other baselines**; Besides, I am not persuaded by the conclusion thats some path can only be captured by your method, e.g., you should try compare with explainable GNN (like GNNExplainer) on that. **From my view, the best way is to give a counterexample, theoretically (not empirically) prove why some temporal path can only captured by your method, not any other baselines.**
> > 2. I think explainability should be a contribution of your method, thus add explainability extraction to your method will improve the score, e.g., **how to extract key/anomalous path from your deep model inference**.
> > 3. As point out also by other authors, scalability is also something to defail. For example, in finance domain, the transaction amount can be a giant number, and building and computing over hypergraph is even more costy, **how to achieve a feasible training and inference on giant temporal graph (at least million nodes) should be discussed**.

---

### Official Review · Reviewer_jhfT · 2024-11-08

**Soundness:** 3
**Presentation:** 2
**Contribution:** 2
**Rating:** 5
**Confidence:** 4

**Summary:**

This paper studies how to model temporal patterns in dynamic graphs and proposes to use statistical graph inference to identify sequence anomalies for graph augmentation and perform message passing on it to capture inductive biases of sequence patterns. The effectiveness of the model is tested on a synthetic dataset and five empirical datasets for static node classification.

**Strengths:**

- The idea of augmenting the input graph for message passing using a statistical null model to detect abnormal temporal patterns and distinguish sequences beyond frequency is interesting.

- The adapted HYPA offers an interpretable way to identify unusual sequences in dynamic graphs, and the proposed HYPA-DBGNN achieves improved performance over baseline models on multiple empirical datasets.

**Weaknesses:**

- The core techniques of using De Bruijn graphs and hypergeometric testing are well established in time series data analysis. The proposed HYPA-DBGNN is, to some extent, an interesting adaptation for GNNs.

- Using De Bruijn graphs with statistical augmentation is a sound approach. However, the paper would benefit from more discussion on why it is optimal for this purpose under the setting for node classification on time-varying graphs, rather than simply improving from DBGNN.

- The evaluation focuses on a limited set of small human interaction networks. Testing on a more diverse set of temporal datasets would better substantiate the model’s broader applicability and generalizability across domains.

**Questions:**

- Q1 The authors state that computational complexity may not be a limiting factor. Could the authors further clarify the complexity increased from DBGNN. How would they compare to standard temporal GNNs? Meanwhile, all datasets used for evaluation have less than 500 nodes, can the proposed method scale to larger graphs?
- Q2 The results in Tabe 1 on synthetic data try to highlight patterns that only high-order models can discern. However, the results are not convincing or interpretable, especially the discussion of the baseline HONEM (even a strong one in Table 2) is very limited.
- Q3 The proposed method claims to have better interoperability by introducing HYPA. Could the authors elaborate more on how it is made more expressive by not relying on the transitivity assumption?

---

> ### Author Response · Authors · 2024-11-20
>
> We thank you for your supportive review and the positive feedback about our work. We have addressed most of your questions in the aggregate response to all reviewers.
>
> .
>
> We additionally like to address why we focus on the given **data sets**:
>
> The choice of data used in our evaluation was based on the need to have a sufficiently large number of observed interactions compared to the number of nodes and edges. This is necessary in order to observe a number of time-respecting paths that is sufficiently large to establish significant deviations from the expected values calculated from the model. Most available large data sets on temporal graphs have large numbers of nodes and edges, but are too sparse in terms of observed time-respecting paths.
>
> .
>
> Regarding the **performance of HONEM**:
>
> The synthetic data sets contain random fluctuations in the possibly higher-order edge frequency statistics due to the randomized creation and splitting. HONEM that heavily utilized higher-order edge frequencies seems to learn these fluctuations and not the pattern because the standard deviation is high and the performance does not increase for the data set with pattern.
> Also for the empirical data HONEM relies on the higher-order edge frequencies. As a result it performs better than first-order methods, especially in  data sets like Hospital or Workplace where the higher-order statistics are linked to the classification task (see Fig. 2).
>
> .
>
> Regarding the **transitivity assumption**:
>
> We indeed encode the transitivity with the edges of the De Bruijn graph. The edge weight is extended by the HYPA scores such that we are able to express the representativeness of the transitive dependencies. We explain the example for the increased expressivity in the general response.

---

> > ### Comment · Reviewer_jhfT · 2024-11-27
> >
> > Thank you for your response. While I appreciate the effort, I feel that some key concerns remain unaddressed.
> >
> > Your explanation on the computational complexity and scalability of HYPA-DBGNN compared to standard temporal GNNs was insufficient (actual runtime comparison and preprocessing cost remain unclear). Additionally, the interpretability of synthetic data results still lacks clarity and convincing explanation. I also believe that the discussion around expressivity and transitivity could benefit from more specific examples with HYPA (besides Fig. 4 in Appx. I).
> >
> > Based on my above reviews and authors' responses, I stand by my initial rating. I encourage further refinement in these areas to improve the manuscript.

---

### Author Response · Authors · 2024-11-20

Thank you for the detailed reviews. We are grateful that the reviewers acknowledge the novelty and importance of our contribution. UJFa appreciates that our work considers “an interesting problem with many industrial applications”. Y5VP highlights the “novel approach” to introduce “De Bruijn graphs into temporal graph analysis” and the “extensive experiments” conducted “to demonstrate the effectiveness of the proposed method”. Also, jhfT finds the “idea of augmenting the input graph [..] using a statistical null model [to] distinguish sequences beyond frequency [...] interesting” and emphasizes that our method “offers an interpretable way to identify unusual sequences in dynamic graphs” that leads to “improved performance over baseline models on multiple empirical datasets”.  We are also pleased that UJFa acknowledged that “the paper is well organized and motivated” and that the experiments “appear to be carefully performed and well motivated” such that the results “provide evidence that the method works”. We also thank the reviewers for the questions and suggestions which we address below.

.

Even though our method already shows “promising performance gains” (UJFa) there is a deeper interest in the expressiveness of the GNN with inductive bias.

We demonstrate the enhanced expressivity with the synthetic data set. A certain class of patterns is encoded in one instance of the synthetic data set. The structure of this pattern is thoroughly explained in appendix I. As described, the second synthetic data set does not contain this pattern but the frequencies match the first one. In the evaluation, we show that no baseline method is able to improve the classification performance on the data set with pattern compared to the performance of the data set without patterns.  Hence, they rely on the edge frequencies that are the same in both data sets. On the other hand our method is able to learn a perfect classification for the data set with pattern. Hence our method is able to express this pattern.

.

Reviewer jhfT asked about the “complexity” and UJfa asked for the “run-time scaling”.

In appendix D, we included a theoretical analysis of the runtime of our method depending on the size of the temporal graph. This analysis actually shows that our model has reasonable runtime even for large data sets. These theoretical bounds are further corroborated by empirical evaluations in Ref. [45], which shows the scalability of the De Bruijn-based methods in larger data sets. They show, as also noted by receiver Y5P, that even though the De Bruijn Graph introduces new edges, it still retains sparsity in empirical data sets, making it no overly more dense than the graphs used by standard GNNs. The additional calculation of the HYPA scores requires a single traversal of the De Bruijn graph and it is done in a pre-processing step. The pre-processing takes only a negligible amount of time compared to the training of the model. To further remove this pre-processing, we provide an ablation study in appendices A and B that contain a simplified score that can be directly calculated dynamically during training.

.

We comment on using additional baseline models:

We agree that there is a range of methods for dynamic node property prediction in temporal graphs (TGAT, DySAT) and even a larger range for static node classification in static graphs (GAT, PINE, …).  However, this work focuses on predicting static node properties, while using patterns in a temporal graph. This limits the choice of suitable baselines that can be used without making major adaptations to the architecture.

However, we include baselines from those other domains to represent those methods. We observe that node classification methods for static tasks miss the temporal information leading to a worse performance. Even though there might be more sophisticated methods in that area they still miss out the crucial information by design.

The methods for dynamic node prediction incorporate temporal information but focus on dynamic changing node classes and not on higher-order dependencies. Hence, major adaptations need to be made to compare those methods.  This is not trivial and deserves its own research which becomes visible by the question of LQNX for another implementation. The evaluation of different ideas, including the proposed one, with respect to the performance led to the presented version.

To conclude, we chose representative candidates from dynamic node prediction and static node classification approaches. What we show in our work is that because the task we propose is novel, and different from what these standard approaches were developed for, these models show weak performances.

.

We address further individual reviewers suggestions in a direct response.

We thank the reviewers for their time and their careful examination of the notation that we revise in the camera ready version. Thanks to the reviewers, we will substantially improve our manuscript.

---

### Meta-Review · Area_Chair_oJmz · 2024-12-17

**Metareview:**

## Summary
The paper proposes HYPA-DBGNN, a method for static node classification on temporal graphs. The model uses De Bruijn graphs to encode sequential patterns and employs a null model correction via hypergeometric testing to identify and adjust for anomalous temporal patterns. Message passing is then performed on the augmented graph to capture inductive biases. Experiments on synthetic and real-world datasets demonstrate performance improvements over baselines.

## Strengths
- Interesting Use of De Bruijn Graphs: The application of De Bruijn graphs for temporal graph augmentation is novel and provides a structured way to model sequential dynamics.
- The introduction of a null model (HYPA) for anomaly detection adds interpretability and effectively distinguishes temporal sequences based on frequency.
- The paper evaluates the method on both synthetic and empirical datasets, showcasing improvements in accuracy over prior approaches.

## Weaknesses
- The method appears to be an incremental extension of DBGNN with null model correction. The need for this correction lacks strong theoretical justification.
- The paper suffers from unclear notations, insufficient explanation of key concepts (e.g., De Bruijn graphs), and missing details relegated to appendices, making it difficult to follow.
- Inadequate Baselines and Experiments: The evaluation excludes recent temporal GNN baselines (e.g., CAWN, TGAT, DySAT) and focuses on small datasets, limiting the scope and impact of the results. Additionally, performance gains over DBGNN are modest, with overlapping confidence intervals.

While the paper introduces a novel application of De Bruijn graphs and a statistical null model for temporal graph augmentation, it lacks sufficient novelty, clarity in presentation, and comprehensive experimental validation. The method’s incremental nature, unclear theoretical motivation, and omission of strong baselines limit its overall contribution and impact.

**Additional Comments On Reviewer Discussion:**

The common major concerns raised by the reviewers are:
- Clarity and Presentation Issues: The paper is difficult to follow due to inconsistent or unclear notations, lack of explanations for key concepts (e.g., De Bruijn graphs), and important details being placed in the appendix. The presentation suffers from typos, unclear mathematical exposition, and missing visual aids (e.g., a framework diagram).

- Limited Novelty and Theoretical Justification: The method appears to be an incremental extension of DBGNN, with the null model correction lacking strong theoretical justification. The rationale for using De Bruijn graphs and why they enhance performance in this specific setting is not adequately explained.

- Insufficient Experiments and Baselines: The experiments are conducted on small datasets and lack comparisons with modern temporal GNN baselines (e.g., CAWN, TGAT, DySAT). Reported performance improvements over DBGNN are modest, with overlapping confidence intervals, raising concerns about the method's impact.

The authors tried to address the above points, but explanation on the computational complexity and scalability of HYPA-DBGNN compared to standard temporal GNNs was insufficient (actual runtime comparison and preprocessing cost remain unclear). Additionally, more experiments are required (e.g., experimental results on real datasets has giant confidence interval and scalability).

---

### Decision · Program_Chairs · 2025-01-22

Reject